# The efficacy and safety of topical wound oxygen therapy for chronic refractory wounds at high altitude: Protocol for a randomized controlled clinical trial

Shuang Lin[1,2], Lihong Chen[1,2], Dawei Chen[1,2], Yun Gao[1,2], Chun Wang[1,2], Xingwu Ran[1,2]*

**1** Department of Endocrinology & Metabolism, and Center for High Altitude Medicine, West China Hospital, Sichuan University, Chengdu, China, **2** Innovation Research Center for Diabetic Foot, Diabetic Foot Care Center, West China Hospital, Sichuan University, Chengdu, China

* ranxingwu@163.com

## Abstract

### Background

The hypobaric and hypoxic characteristic of plateau regions can lead to sustained hypoxia, inadequate tissue perfusion, and an inflammatory response at wounds sites. Topical wound oxygen therapy (TOT) shows potential in elevating transcutaneous oxygen tension, stimulating the production of vascular endothelial growth factor, promoting angiogenesis and tissue remodeling, thereby accelerating wound healing. However, there is a scarcity of research investigating the therapeutic efficacy of TOT for treating chronic refractory wounds in long-term inhabitants of plateau areas. The present study aims to evaluate the efficacy and safety of TOT as a viable treatment option for chronic refractory wounds in individuals residing in high-altitude environments. Positive findings could establish TOT as a valuable and safe complementary strategy for managing such wounds in high-altitude settings.

### Methods and analysis

This study is a randomized controlled clinical trial involving 250 patients residing at high altitudes. Participants will be randomly assigned to receive either TOT or sham oxygen therapy over a 12-week period, followed by a one-year follow-up. In addition to the experimental interventions, all participants will receive standard care for their chronic wounds. The primary outcome measure is the wound healing rate at the end of the 12-week intervention. Secondary endpoints include the reduction in ulcer area after intervention, the time required for ulcer healing, the recurrence rate of ulcers, the amputation rate, pain levels, and any adverse events.

**Data availability statement:** No datasets were generated or analysed during the current study.

**Funding:** XR received the funding from the Health Commission of Sichuan Province (Grant No. 23LCYJ 042), 1.3.5 Project of Center for High Altitude Medicine, West China Hospital, Sichuan University (Grant No. GYYX 24002), 1.3.5 Project for disciplines of excellence, West China Hospital of Sichuan University (Grant No. ZYGD24005), Sichuan Science and Technology Program (Grant No. 2024 YFFK0290), and Science and Technology Bureau of Sichuan Province (Grant No. 24ZDYF0679). The funders had no role in study design, data collection and analysis, decision to publish, or preparation of the manuscript.

**Competing interests:** NO authors have competing interests.

## Trial registration

The trial has been officially registered in the Chinese Clinical Trial Registry, and assigned the registration number ChiCTR-2400083602.

## Background

Chronic wounds can be attributed to a variety of factors, such as diabetic ulcers, infectious lesions, pressure sores, and traumatic injuries [1]. The management of chronic wounds poses a substantial challenge in contemporary medical practice, owing to their multifactorial etiology, extended treatment duration, significant financial burden, and propensity for recurrence and disability [2]. In high-altitude regions, particular attention has been directed towards addressing refractory chronic wounds within the framework of plateau medicine.

Due to the inherent low-pressure and hypoxic conditions of high-altitude plateaus, combined with persistent tissue hypoxia and decreased perfusion at the wound site, chronic wounds in these regions display distinct pathophysiological characteristics compared to those in plains [3]. A retrospective study conducted at the Qinghai Provincial People's Hospital indicated that among the elderly residing on the plateau, diabetic foot was the leading cause of chronic wounds, accounting for 51.9% of all cases, followed by pressure sores, post-surgical infections, traumatic ulcers, venous ulcers, and arterial ulcers [4]. The microbial profiles and antibiotic resistance patterns associated with infections in plateau wounds markedly differ from those observed in plains [5]. Chronic wounds in high-altitude settings are especially susceptible to complications such as fat necrosis, incisional infection, wound hematomas, and delayed healing [6]. Furthermore, comorbidities such as chronic cardiopulmonary diseases, which are more prevalent in plateau regions, can exacerbate the challenges associated with non-healing wounds.

However, the recommendations for caring for chronic wounds at high altitudes are not substantially distinct from those in plains. Limited literature has been published to investigate interventions that could enhance wound healing in high-altitude environments. In a study, Zhao et al showed that a plateau refractory chronic wound may heal after platelet lysate gel treatment [7]. Standard of care for chronic wounds includes off-loading, anti-infection strategies, debridement, local wound care, negative-pressure wound therapy, and the application of various dressings. Interventions tailored to address the unique pathophysiology of wounds at high altitudes may offer promising avenues for improving wound healing outcomes.

Oxygen serves as a vital component in the wound healing process, assuming a crucial role at each stage. During the inflammatory phase, the production of reactive oxygen species (ROS) activates phagocytes to eradicate pathogens, inhibits microbial proliferation, and aid in the clearance of necrotic tissue. In the proliferation phase, oxygen plays a crucial part in collagen synthesis, the formation of the extracellular matrix, and angiogenesis. Localized tissue hypoxia poses a substantial obstacle to efficient wound healing [8]. Our prior studies have demonstrated that hypoxia can result in the dysregulation of hypoxia-inducible factor (HIF-2α), leading to delayed wound healing

[9]. Factors contributing to chronic tissue hypoxia encompass vasoconstriction induced by inflammation, edema, and pain; compromised limb blood supply due to peripheral artery disease; and heightened oxygen demand caused by wound infection. In plateau regions, the hypobaric and hypoxic conditions may further intensify chronic tissue hypoxia.

Oxygen has been utilized in clinical settings for wound healing since the 1960s. Oxygen supplication to the wound have gained much attention in biomaterial research and development and showed promising therapeutic prospect for wound healing at high altitudes [10,11]. However, there have been a long way for these biomaterials to become in practice. A systematic review indicated that hyperbaric oxygen therapy (HBOT) can significantly improve wound healing and decrease minor and major amputation in individuals with diabetic foot ulcers [12]. However, several limitations impede the widespread adoption of HBOT, including restricted availability, specific contraindications, and logistical challenges associated with patient transportation.

To address these deficiencies, TOT has been introduced. Preclinical animal studies have demonstrated that this therapy can increase transcutaneous oxygen partial pressure, promote vascular endothelial growth factor expression, enhance angiogenesis and tissue remodeling, and accelerate wound healing [13–15]. Clinical controlled trials have shown that TOT significantly improves the ulcer healing rate at 12 weeks (76% vs. 46%, P<0.001) [16] and reduces the time required for complete wound healing(56 days vs. 93 days) [17]. Results from a global multicenter randomized double-blind controlled trial showed that TOT significantly enhances the rate of wound healing (Odds ratio, OR 6, 97.8% confidence interval, CI 1.44–24.93, P=0.004) [18]. Specifically, while some studies suggest the efficacy of TOT therapy in wound healing, others report inconsistent results. Notably, the studies by Driver et al. and He et al. did not demonstrate any additional benefits of TOT therapy for wound healing compared to moist wound therapy [19,20]. A meta-analysis published in 2024 showed that TOT therapy can significantly increase wound healing (RR 1.77, 95%CI 1.18–2.64, P=0.005) [21]. However, there remains insufficient evidence regarding the efficacy of TOT for chronic refractory wounds in patients residing in high-altitude regions.

The primary objective of this research project was to evaluate the efficacy and safety of TOT in managing chronic refractory wounds among patients residing in plateau regions, employing a randomized controlled study design. This study aimed to provide a feasible and safe therapeutic option for addressing the prevalent issue of chronic and refractory wounds in plateau settings.

## Methods and design

### Objective

The primary objective of this study was to evaluate the efficacy of TOT in managing chronic refractory wounds among patients residing in high-altitude regions through randomized controlled trials.

### Research design

The present study is designed as a randomized, double-blinded, placebo-controlled clinical trial and has received approval from the Ethics Committee on Clinical Trial at West China Hospital of Sichuan University (Approval No. 2023(2291)). The study will adhere to the principles outlined in the Declaration of Helsinki and the Guidelines for Good Clinical Practice. Written informed consent will be obtained from all participants prior to participation. This trial has been registered with the Chinese Clinical Trial Registry (Registration Number: ChiCTR-2400083602). In addition to receiving standard wound care, participants will be randomly allocated to either the TOT group or the sham therapy group. A schematic diagram outlining the design of this clinical trial is presented in Supporting information S1 File.

### Study population

The recruitment of patients with chronic refractory wounds from high-altitude regions will be conducted at West China Hospital, Sichuan University. All individuals seeking treatment for chronic wounds in high-altitude areas will undergo a

comprehensive and rigorous screening process based on predefined inclusion and exclusion criteria. Following the acquisition of informed consent, eligible patients with chronic refractory wounds from these regions will be enrolled in the study. This trial is scheduled to commence in Oct. 2024 and is expected to conclude by June 2026.

**Inclusion criteria.** ① Participants must be aged between 18 and 80 years; ② Participants are required to reside in regions with an elevation of no less than 2,500 meters above sea level; ③ Chronic refractory wounds are defined as those that have not healed within a four-week period, including chronic diabetic skin ulcers and venous insufficiency ulcers of the lower limbs; ④ Adequate blood flow is required, as evidenced by an Ankle-Brachial Index (ABI) of 0.6 or higher, or a transcutaneous oxygen tension exceeding 30 mmHg; ⑤ The ulcer area must measure between 1 and 20 cm$^2$.

**Exclusion criteria.** A potential subject who fulfills any of the following criteria will be excluded from the trial: ① Limb gangrene; ② Osteomyelitis; ③ Malignant neoplasm; ④ HIV-positive status; ⑤ Severe cardiovascular, hepatic, renal, respiratory, or neurological disorders, as well as other systemic diseases; ⑥ Long-term use of corticosteroids or other immunosuppressive agents; ⑦ Pregnant women, or those planning to conceive within three months prior to or following the initiation of treatment, as well as lactating women; ⑧ individuals with psychiatric conditions, significant cognitive impairments, substance abuse disorders (including alcohol and drugs dependencies), and an inability to adhere to the treatment regimen.

## Withdrawal of individual subjects

Participants retain the right to voluntarily withdraw from the trial at any time for any reason. Otherwise, it is advised that participants cease their participation under the following circumstances: (1) the emergence of serious adverse events necessitating a temporary interruption in treatment; (2) inadequate progress in ulcer healing, defined by less than a 20% reduction in ulcer size following four weeks of therapy; (3) the onset of a critical clinical infection requiring urgent surgical intervention; and (4) the appearance of indications for major toe or limb amputation in the affected extremities.

## Informed consent

The acquisition of written informed consent is mandatory for all participants. In cases where participants are unable to provide consent, it must be obtained from their legally designated representatives.

## Sample size calculation

The study conducted by Blackman demonstrated that the 12-week ulcer healing rate in the standard treatment group for diabetic foot ulcers was 42.8%, whereas the wound healing rate in the group receiving TOT was significantly higher at 85.2% [17]. Our previous research documented a wound healing rate of 69% in the standard treatment group for diabetic foot ulcers [22]. Assuming a baseline wound healing rate of 60% in the standard treatment group over a 12-week period, it is hypothesized that the incorporation of TOT would result in a 30% relative improvement in the wound healing rate, i.e., from 60% to 78%. This hypothesis will be evaluated using a two-sided test with a significance level (α) of 0.05 and a statistical power (1-β) of 0.8. Sample size estimation, conducted using PASS 15 software and based on the methodology for comparing two-group proportions in a randomized controlled trial, indicated that each experimental and control group should comprise approximately 100 participants, totaling 250 participants when accounting for an expected dropout rate of approximately 20%.

## Randomization and allocation concealment

The eligible participants will be randomly assigned in a 1:1 ratio to either receive TOT or sham oxygen therapy. Additionally, all subjects will receive standard wound care as prescribed by their attending physician. The random sequence will be generated by a researcher utilizing R software with the simple randomization method. A designated technician will oversee

the sequence numbers and monitor the treatment procedures. Both the attending physician and participants will remain blinded to the specific oxygen therapy protocol.

### Interventions

Following randomization, all participants will be assigned to either the treatment group, which will receive TOT, or the control group, which will undergo sham therapy. Additionally, all patients will receive standard medical care for chronic wounds, which include the management of blood sugar, blood pressure, and lipid profiles; antiplatelet aggregation and anticoagulation therapies; infection prevention measures; promotion of physical activity; pressure relief interventions; and debridement procedures. The application of platelet-rich gel and negative pressure wound therapy will also be incorporated. To ensure that all patients received standard wound care, a checklist with the abovementioned contents will be used in the study.

Treatment Group: In addition to receiving standard care for chronic wounds, patients will undergo TOT at their bedside within the ward. This therapy will be delivered using the lower limb wound oxygen therapy device (G00001) supplied by Odie Oxygen Therapy Company, based in Ireland. The treatment involves the delivery of pressurized oxygen, with a pressure range of 0–50 mbar, generated by an oxygen concentrator operating at a flow rate of 10 L/min. Each session will last 90 minutes and will be conducted once daily, five days per week.

Control Group: Patients in the control group received standard wound care and sham therapy. The sham devices looked and operated identically with the active TOT therapy, with the sole difference being the use of ambient air as the gas source instead of an oxygen supply.

### Outcomes

**Primary outcomes.** The wound healing rate at 12 weeks: the proportion of participants achieving complete wound healing, defined as full epithelialization of the wound within a 12-week timeframe.

**Secondary outcome.**

1. Rate of ulcer area reduction: percent change in wound size after 12 weeks of treatment

2. Time to ulcer healing: duration until complete wound closure

3. One-year ulcer recurrence rate: proportion of ulcers that recur within one-year post-healing

4. Amputation rate at 12 weeks: incidence of amputation within the first 12 weeks

5. VAS score: Visual analogue scale for pain assessment

6. Wound-QoL: Questionnaire assessing quality of life in patients with chronic wounds

### Follow-up

Patients will be randomly allocated to either the treatment group or the control group and will receive the respective interventions until complete wound healing occurs or for a maximum duration of 12 weeks. Digital photography will be employed to systematically document images on a weekly basis. The researchers will determine whether the outcomes were reached. Because of the objective feature, the outcome assessment for wound healing will not be blinded or centralized. The Image J software will be utilized to quantitatively measure the area of the foot ulcer during each debridement and dressing change procedure. The total follow-up period will span one year, with evaluations conducted at 4-, 8-, 12-, 24-, and 52-weeks post-randomization to assess wound healing, ulcer recurrence, and the incidence of amputation. The items to be measured and the time window of data collection are shown in Fig 1.

| | STUDY PERIOD | | | | | | |
|---|---|---|---|---|---|---|---|
| | Enrolment | Allocation | Post-allocation | | | | Close-out |
| TIMEPOINT | | 0 | 4w | 8w | 12w | 24w | 52w |
| **ENROLMENT:** | | | | | | | |
| Eligibility screen | X | | | | | | |
| Informed consent | X | | | | | | |
| Allocation | | X | | | | | |
| **INTERVENTIONS:** | | | | | | | |
| Topical wound oxygen therapy | | | •————————• | | | | |
| placebo | | | | | | | |
| **ASSESSMENTS:** | | | | | | | |
| Wound healing rate | | | | | X | | |
| ulcer area reduction | | | | | X | | |
| Amputation rate | | | | | X | | |
| Time to ulcer healing | | | X | X | X | X | X |
| Ulcer recurrence rate | | | | | | | X |
| VAS score | | X | | | X | | X |
| Wound-QoL | | X | | | X | | X |

**Fig 1. The schedule of enrolment, interventions, and assessments.**

## Adverse events

The incidence of all adverse events, whether reported by participants or observed by the research investigator, will be meticulously documented in accordance with the study protocol. Appropriate interventions will be promptly implemented in response to the severity of each adverse event. In the case of serious adverse events, it is imperative to notify both the expert committee and the ethics committee within 24 hours.

## Data collection and management

The principal investigator (PI) is responsible for the design, implementation and management of the study. Other researchers are tasked with clinical diagnosis, participant screening, obtaining informed consent, administering interventions, collecting clinical samples, conducting follow-up, and ensuring data entry. A researcher will use a case report form (CRF) to collect data. A clinical trial associate will verify whether the data filled in the CRF are complete and accurate. All the personal information of subjects collected during this clinical trial will be remained strictly confidential. The data collected in the study will be stored in the hospital. The primary investigator has the access to the final dataset. The dataset without identifiable information will be available from the corresponding author on reasonable request.

## Statistical analysis

R software will be utilized to conduct statistical analyses. The multiple imputation approach will be utilized to address missing data. Statistical analysis will be conducted using an intention-to-treat approach. Specifically, the chi-square test will be utilized to evaluate the primary outcome measure, namely the wound healing rate at 12 weeks. For sparse data, Fisher's exact test will be used. Logistic regression analysis will be conducted to investigate potential confounding factors that may influence wound healing. Confounding factors, such as sex, age, history of diabetes, ulcer categories, wound area, and infection status, will be adjusted to ensure the reliability of the findings. Subgroup analysis will be conducted according to age, wound categories, and wound area. The reduction rate of ulcer area and wound healing time at 12 weeks were evaluated using either the independent t-test or the Mann-Whitney, contingent upon meeting the assumptions for parametric analysis. Furthermore, the chi-square test will be applied to assess the ulcer recurrence rate and the amputation rate. Kaplan-Meire survival analysis will be performed to compare wound healing between the two groups, with the log-rank test used for statistical comparison. Unadjusted and adjusted Cox proportional hazards models were employed to evaluate the association between TOT therapy and wound healing. Multivariable models were adjusted for covariates such as age, sex, history of diabetes, ulcer categories, wound area, and infection status. A significance level of $P < 0.05$ will be considered statistically significant.

## Patient and public involvement

The formulation of the research question, the determination of outcome measures, and the design of the protocol were conducted without the involvement of patients or members of the general public.

## Discussion

Chronic wounds represent a substantial public health challenge on a global scale. The high-altitude environment exacerbates the complexity of the wound healing process. In order to improve wound healing outcomes, various interventions are currently being developed and evaluated in clinical settings. This trial has the potential to provide definite evidence that TOT is an efficacious and safe therapeutic option for patients with chronic wounds in high-altitude regions.

In fact, the efficacy of TOT in promoting the healing of diabetic foot ulcers has been substantiated by a randomized clinical trial [18]. A systematic review and meta-analysis further support these findings, indicating that TOT is a viable treatment option for chronic Wagner 1 or 2 diabetic foot ulcers in the absence of infection and ischemia, with a risk ratio of 1.59 (95%CI 1.07–2.37; $p = 0.021$) [23]. However, it is crucial to acknowledge that previous studies have predominantly focused on the effectiveness of this therapy in mild cases. To date, no research has specifically examined its application in high-altitude populations. The results of this trial may provide valuable insights into the potential benefits of TOT in such settings.

In this study, the standard of care for chronic wounds is established by the attending physicians and is not governed by a standardized protocol. This methodology reflects actual clinical practice. Consequently, it is reasonable to assume that the results can be extrapolated to the broader Chinese population inhabiting high-altitude regions. However, it should be emphasized that the implementation of TOT therapy requires the availability of TOT device, which may pose challenges in resource-limited settings. In fact, the procedure for TOT therapy is relatively straightforward and can be effectively performed by healthcare practitioners following minimal training. Nevertheless, the single-center design of this study may impose limitations on its external validity. Further multi-center studies could potentially strengthen the generalizability and reliability of these findings. Of course, it should be noted that it may be a challenge to recruit enough subjects due to the lower population density at high altitudes.

In conclusion, the results of this trial will provide further evidence to substantiate the efficacy of TOT in the treatment of chronic wounds in high-altitude regions. Further studies may be conducted to elucidate the underlying mechanisms of TOT therapy and assess its applicability across diverse wound categories.

## Trial status

This protocol constitutes the preliminary version, which was officially approved on March 13, 2024. Recruitment of participants is scheduled to commence in October 2024. The study is expected to reach its conclusion in June 2026. The findings from this study will be disseminated through publication in relevant, peer-reviewed scientific journals by the research team. In addition, the research team will actively communicate the findings of this study with healthcare providers and policymakers at academic conferences.

## Supporting information

**S1 File. Flowchart of the Clinical Trial** .
(PDF)

**S2 File. SPIRIT Checklist for Trials** .
(DOCX)

**S3 File. Study protocol approved by your ethics committee** .
(PDF)

**S4 File. Study protocol translated** .
(PDF)

## Acknowledgments

Not applicable.

## Author contributions

**Conceptualization:** Shuang Lin, Lihong Chen, Chun Wang, Xingwu Ran.

**Methodology:** Shuang Lin, Lihong Chen, Dawei Chen, Yun Gao, Chun Wang.

**Supervision:** Dawei Chen, Yun Gao, Chun Wang.

**Writing – original draft:** Shuang Lin, Lihong Chen.

**Writing – review & editing:** Xingwu Ran.

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
