## [Decision Letter · Decision Letter 0]

PONE-D-25-21928The efficacy and safety of topical wound oxygen therapy for chronic refractory wounds at high altitude: protocol for a randomized controlled clinical trialPLOS ONE

Dear Dr. Ran,

Thank you for submitting your manuscript to PLOS ONE. After careful consideration, we feel that it has merit but does not fully meet PLOS ONE’s publication criteria as it currently stands. Therefore, we invite you to submit a revised version of the manuscript that addresses the points raised during the review process.

We look forward to receiving your revised manuscript.

Kind regards,

Nan Jiang

Academic Editor

PLOS ONE

Reviewers' comments:

Reviewer's Responses to Questions

**Comments to the Author**

1. Does the manuscript provide a valid rationale for the proposed study, with clearly identified and justified research questions?

Reviewer #1: Yes

Reviewer #2: Partly

Reviewer #3: Yes

Reviewer #4: Yes

2. Is the protocol technically sound and planned in a manner that will lead to a meaningful outcome and allow testing the stated hypotheses?

Reviewer #1: Yes

Reviewer #2: Partly

Reviewer #3: Yes

Reviewer #4: Partly

3. Is the methodology feasible and described in sufficient detail to allow the work to be replicable?

Reviewer #1: Yes

Reviewer #2: Yes

Reviewer #3: Yes

Reviewer #4: No

4. Have the authors described where all data underlying the findings will be made available when the study is complete?

Reviewer #1: Yes

Reviewer #2: Yes

Reviewer #3: Yes

Reviewer #4: No

5. Is the manuscript presented in an intelligible fashion and written in standard English?

Reviewer #1: Yes

Reviewer #2: Yes

Reviewer #3: Yes

Reviewer #4: Yes

6. Review Comments to the Author

You may also provide optional suggestions and comments to authors that they might find helpful in planning their study.

Reviewer #1: The efficacy and safety of topical wound oxygen therapy for chronic refractory wounds at high altitude: protocol for a randomized controlled clinical trial

Study Protocol

Generally:

This study is well-structured and scientifically sound, incorporating an appropriate methodology to assess TOT in high-altitude wound care. To strengthen its impact, additional consideration of alternative treatments, environmental confounders, accessibility concerns, and subgroup analyses is recommended.

Abstract:

Strengths:

The title is clear and directly conveys the study’s focus. The abstract concisely summarizes the rationale, methodology, and expected impact, making it accessible to readers.

Comments/Recommendations:

The abstract could briefly highlight potential limitations to set appropriate expectations for the study's findings.

Consider incorporating a sentence acknowledging challenges related to study execution, such as patient adherence or environmental variability.

see attached file

Reviewer #2: This study addresses an important and underexplored clinical problem — the management of chronic refractory wounds in high-altitude populations using topical wound oxygen therapy (TOT). The proposed randomized, double-blinded, placebo-controlled design is methodologically robust, and the outcomes chosen are relevant and clinically significant. However, several key areas require revision to improve the protocol's scientific rigor, reproducibility, and applicability:

1. Wound Type Stratification:

The inclusion of multiple chronic wound types (e.g., diabetic foot ulcers, venous ulcers) without subgroup stratification or analysis is a significant limitation. Different wound etiologies have distinct pathophysiology and may respond differently to oxygen therapy. The authors should consider stratified randomization or include planned subgroup analyses.

2. Standard of Care Consistency:

"Standard wound care" is left to physician discretion without a defined protocol. This introduces variability and may affect the internal validity of the study. A minimum standard wound care protocol should be outlined (e.g., dressing selection, debridement practices, offloading strategies) to ensure consistency across groups.

3. Blinding of Sham Therapy:

The manuscript lacks detail on how sham oxygen therapy (ambient air) will be made indistinguishable from the active TOT, which could compromise blinding. The authors should clarify how the control setup will simulate the sensory experience of TOT (e.g., pressure, temperature, sound).

4. Patient-Centered Outcomes Missing:

Pain and quality of life are briefly mentioned but not evaluated using standardized tools. These outcomes are crucial in chronic wound studies. The authors are encouraged to incorporate validated instruments such as the Visual Analog Scale (VAS) for pain and the EQ-5D or Wound-QoL for quality of life.

5. Literature Review Imbalance:

The background heavily emphasizes positive studies supporting TOT, with little discussion of conflicting or negative findings. A more balanced review of current literature is necessary to frame the rationale objectively.

6. Generalizability and Limitations:

While the setting is appropriate for the study goals, the single-center design limits broader applicability. This should be acknowledged explicitly in the discussion, with a recommendation for future multi-center research.

7. Microbiological Context:

Given the noted differences in microbial profiles in high-altitude wounds, the protocol could be strengthened by including infection tracking or microbial analysis as exploratory endpoints.

8. Figures and Visuals:

Figures 1 and 2 are informative but visually dense. Consider simplifying and clarifying their design for better comprehension.

In conclusion, this is a promising and relevant study. With moderate revisions addressing the points above, it can make a valuable contribution to wound care research in resource-limited, high-altitude environments.

Recommendation: Revision Required Before Acceptance

Reviewer #3: In The efficacy and safety of topical wound oxygen therapy for chronic refractory wounds at high altitude: protocol for a randomized controlled clinical trial, Lin et al. describe a well thought out protocol for a clinical evaluation for TOT therapy. The authors do a good job explaining the rationale and need for the study. The experimental design, inclusion and exclusion criteria, and evaluation plan seem appropriate for the study. Overall, this protocol is appropriate for publication with minor revisions.

Background

- Briefly introduce the basic mechanism of TOT therapy

Methods

- Briefly describe the rationale behind therapy regimen. I.e. is the timing and O2 concentration based upon animal trials? Are there any considerations for implementing in human patients?

Reviewer #4: The primary objective of this randomized, double-blind, placebo-controlled clinical trial involving 250 residents living at high altitudes is to assess the efficacy and safety of topical wound oxygen therapy (TOT) for treating chronic refractory wounds. Participants will be randomly assigned to receive either TOT or sham oxygen therapy over a 12-week period, followed by a one-year follow-up. The primary endpoint is the wound healing rate at the end of the 12-week intervention. Secondary outcomes include reduction in ulcer area, time to complete healing, ulcer recurrence rate, amputation rate, pain levels, and the incidence of adverse events. If proven effective, TOT could serve as a valuable and safe adjunctive treatment for managing chronic wounds in high-altitude settings. The protocol covers all key aspects of study design, randomization, and statistical issues; however, there are several major statistical concerns about the current manuscript.

Statistical critiques:

1. Although the authors discuss power analysis and sample size estimation, they should explicitly clarify the following: (1) the statistical test used to determine the required sample size, (2) whether a one-sided or two-sided test was applied, and (3) that the reported 30% improvement refers to a relative, not absolute, increase in the wound healing rate. Specifically, the authors should state: “Assuming a baseline wound healing rate of 60% in the standard treatment group over a 12-week period, it is hypothesized that the addition of TOT will result in a 30% relative improvement in healing, i.e., 60% vs. 78%.”

2. Although the manuscript discusses randomization, the authors should explicitly specify the method used—for example, simple randomization (not recommended) or stratified permuted block randomization (highly recommended). If stratified permuted block randomization is chosen, the authors should clearly detail the stratification factor(s), block size, and other relevant parameters in the manuscript.

3. The statistical analysis plan covers the major outcomes but lacks essential methodological clarity and precision. While using the chi-square test for comparing wound healing rates is reasonable, the authors should clarify whether assumptions will be assessed and whether Fisher’s exact test will be used for sparse data. The proposed use of logistic regression to adjust for confounders is appropriate, but the model’s covariates, selection criteria, and diagnostics are not specified. Additionally, time-to-healing is better analyzed with survival methods rather than the proposed t-test, and the mention of Kaplan-Meier analysis is vague—for example, there is no indication of what time-to-event outcome is being evaluated or whether log-rank tests or Cox models will be used.

4. In summary, several important elements are missing from the current statistical analysis plan. There is no mention of how missing data will be addressed (a formal multiple imputation approach is recommended), whether analyses will follow the intention-to-treat principle, or how multiple testing across secondary endpoints will be controlled (either Bonferroni adjustment or false discovery rate [FDR] should be applied). The t-tests for ulcer area and healing time assume no severe skewness (skewness < 1), which should be assessed; otherwise, non-parametric methods such as the Wilcoxon rank-sum test should be used. To improve the rigor of the plan, the authors should clarify test assumptions, specify covariates, define outcomes more precisely, and incorporate appropriate strategies for handling missing data and multiplicity.

7. PLOS authors have the option to publish the peer review history of their article (what does this mean?). If published, this will include your full peer review and any attached files.

Reviewer #1: No

Reviewer #2: **Yes: **Abdulrahman Abdullah Almalki

Reviewer #3: No

Reviewer #4: No

---

## [Author Response · Author response to Decision Letter 1]

2 Jun 2025

Dear Editor-in-Chief,

We extend our gratitude to the Editor and Reviewers for their valuable time and constructive feedback. We highly appreciate the opportunity to resubmit our manuscript following the journal’s requirements. In response, we have thoroughly addressed all the comments raised by the Journal. Additionally, our detailed point-by-point replies to the Reviewers’ comments are provided in the Response to Reviewers file. Tracked changes are provided in the Revised Manuscript.

We look forward to hearing from you at your earliest convenience.

Yours sincerely,

Xingwu Ran

---

## [Decision Letter · Decision Letter 1]

PONE-D-25-21928R1The efficacy and safety of topical wound oxygen therapy for chronic refractory wounds at high altitude: protocol for a randomized controlled clinical trialPLOS ONE

Dear Dr. Ran,

Thank you for submitting your manuscript to PLOS ONE. After careful consideration, we feel that it has merit but does not fully meet PLOS ONE’s publication criteria as it currently stands. Therefore, we invite you to submit a revised version of the manuscript that addresses the points raised during the review process.

We look forward to receiving your revised manuscript.

Kind regards,

Nan Jiang

Academic Editor

PLOS ONE

Journal Requirements:

Reviewers' comments:

Reviewer's Responses to Questions

**Comments to the Author**

1. Does the manuscript provide a valid rationale for the proposed study, with clearly identified and justified research questions?

Reviewer #2: Yes

Reviewer #4: Yes

2. Is the protocol technically sound and planned in a manner that will lead to a meaningful outcome and allow testing the stated hypotheses?

Reviewer #2: Yes

Reviewer #4: Yes

3. Is the methodology feasible and described in sufficient detail to allow the work to be replicable?

Reviewer #2: Yes

Reviewer #4: Yes

4. Have the authors described where all data underlying the findings will be made available when the study is complete?

Reviewer #2: Yes

Reviewer #4: Yes

5. Is the manuscript presented in an intelligible fashion and written in standard English?

Reviewer #2: Yes

Reviewer #4: Yes

6. Review Comments to the Author

You may also provide optional suggestions and comments to authors that they might find helpful in planning their study.

Reviewer #2: This is a resubmission of a protocol I previously reviewed (PONE-D-25-21928). The authors have addressed the major concerns raised in the initial review, particularly regarding blinding clarification, co-intervention standardization, and ethical transparency. The revised protocol is clearer and more robust overall.

The study protocol presents a novel and clinically significant investigation into the use of topical wound oxygen therapy (TOT) for chronic refractory wounds in high-altitude populations. The rationale is well-articulated, supported by both pathophysiological and epidemiological arguments specific to plateau environments, where wound healing is uniquely compromised due to hypoxia and limited perfusion.

The research design is strong, utilizing a randomized, double-blind, placebo-controlled approach with adequate sample size justification and meaningful primary and secondary endpoints. The use of a sham-controlled group and consistent standard care protocols (including wound off-loading, debridement, and infection control) reflects good clinical trial standards. The methodology is reproducible and clearly described.

Minor suggestions include:

Enhancing standardization in how “standard wound care” will be monitored or audited across patients.

Clarifying if outcome adjudicators for wound healing will be blinded or centralized.

Confirming the data repository platform intended for post-trial public data availability.

This trial, once completed, could significantly contribute to the evidence base for wound care in geographically and physiologically challenging environments. The planned 1-year follow-up is a strength and will support long-term safety and efficacy analysis.

Reviewer #4: The authors have responded well to the statistical issues raised in the previous review. There is no further statistical concern about this revised manuscript.

7. PLOS authors have the option to publish the peer review history of their article (what does this mean?). If published, this will include your full peer review and any attached files.

Reviewer #2: **Yes: **Abdulrahman Almalki

Reviewer #4: No

---

## [Author Response · Author response to Decision Letter 2]

18 Jun 2025

Dear editor,

We thank the Editor and Reviewers for their valuable time and constructive feedback. We have reviewed the reference list and we are sure that there were no papers that have been retracted. And we also make some revisions to ensure it is complete and correct. The detailed point-by-point replies to the Reviewers’ comments are provided below.

Reviewer #2:

Minor suggestions include:

Enhancing standardization in how “standard wound care” will be monitored or audited across patients.

Clarifying if outcome adjudicators for wound healing will be blinded or centralized.

Confirming the data repository platform intended for post-trial public data availability.

Response: Thank you for your suggestion. We have made corresponding revisions. Because the outcomes of wound healing were objective, the outcome adjudicators for wound healing will not be blinded or centralized. The data collected in the study will be stored in the hospital. The dataset will be available from the corresponding author on reasonable request.

“To ensure that all patients received standard wound care, a checklist with the abovementioned contents will be used in the study.”

“The researchers will determine whether the outcomes were reached. Because of the objective feature, the outcome assessment for wound healing will not be blinded or centralized.”

“The data collected in the study will be stored in the hospital. The primary investigator has the access to the final dataset. The dataset without identifiable information will be available from the corresponding author on reasonable request.”

Reviewer #4: The authors have responded well to the statistical issues raised in the previous review. There is no further statistical concern about this revised manuscript.

Response: Thank you.

---

## [Decision Letter · Decision Letter 2]

The efficacy and safety of topical wound oxygen therapy for chronic refractory wounds at high altitude: protocol for a randomized controlled clinical trial

PONE-D-25-21928R2

Dear Dr. Ran,

We’re pleased to inform you that your manuscript has been judged scientifically suitable for publication and will be formally accepted for publication once it meets all outstanding technical requirements.

Kind regards,

Nan Jiang

Academic Editor

PLOS ONE

Reviewers' comments:

Reviewer's Responses to Questions

**Comments to the Author**

1. Does the manuscript provide a valid rationale for the proposed study, with clearly identified and justified research questions?

Reviewer #4: Yes

2. Is the protocol technically sound and planned in a manner that will lead to a meaningful outcome and allow testing the stated hypotheses?

Reviewer #4: Yes

3. Is the methodology feasible and described in sufficient detail to allow the work to be replicable?

Reviewer #4: Yes

4. Have the authors described where all data underlying the findings will be made available when the study is complete?

Reviewer #4: Yes

5. Is the manuscript presented in an intelligible fashion and written in standard English?

Reviewer #4: Yes

6. Review Comments to the Author

You may also provide optional suggestions and comments to authors that they might find helpful in planning their study.

Reviewer #4: The authors have responded well to the statistical issues raised in the previous review. There is no further statistical concern about this revised manuscript.

7. PLOS authors have the option to publish the peer review history of their article (what does this mean?). If published, this will include your full peer review and any attached files.

Reviewer #4: No

---

## [Editor Report · Acceptance letter]

PONE-D-25-21928R2

PLOS ONE

Dear Dr. Ran,

I'm pleased to inform you that your manuscript has been deemed suitable for publication in PLOS ONE. Congratulations! Your manuscript is now being handed over to our production team.

Kind regards,

on behalf of

Dr. Nan Jiang

Academic Editor

PLOS ONE